# Global Warming by Geothermal Heat from Fracking: Energy Industry’s Enthalpy Footprints

**DOI:** 10.3390/e24091316

**Published:** 2022-09-19

**Authors:** Leslie V. Woodcock

**Affiliations:** Department of Physics, Faculty of Science and Technology, University of Algarve, 8005-139 Faro, Portugal; lvwoodcock@ualg.pt

**Keywords:** global warming, fracking, atmospheric thermodynamics, geothermal energy, troposphere, radiation balance, enthalpy footprint, entropy footprint

## Abstract

Hypothetical dry adiabatic lapse rate (DALR) air expansion processes in atmosphere climate models that predict global warming cannot be the causal explanation of the experimentally observed mean lapse rate (approx.−6.5 K/km) in the troposphere. The DALR hypothesis violates the 2nd law of thermodynamics. A corollary of the heat balance revision of climate model predictions is that increasing the atmospheric concentration of a weak molecular transducer, CO_2_, could only have a net cooling effect, if any, on the biosphere interface temperatures between the lithosphere and atmosphere. The greenhouse-gas hypothesis, moreover, does not withstand scientific scrutiny against the experimental data. The global map of temperature difference contours is heterogeneous with various hotspots localized within specific land areas. There are regional patches of significant increases in time-average temperature differences, (∆<T>) = 3 K+, in a ring around the arctic circle, with similar hotspots in Brazil, South Africa and Madagascar, a 2–3 K band across central Australia, SE Europe centred in Poland, southern China and the Philippines. These global-warming map hotspots coincide with the locations of the most intensive fracking operational regions of the shale gas industry. Regional global warming is caused by an increase in geothermal conductivity following hydraulic fracture operations. The mean lapse rate (d<T>/dz)_z_ at the surface of the lithosphere will decrease slightly in the regions where these operations have enhanced heat transfer. Geothermal heat from induced seismic activity has caused an irreversible increase in enthalpy (H) input into the overall energy balance at these locations. Investigating global warming further, we report the energy industry’s enthalpy outputs from the heat generated by all fuel consumption. We also calculate a global electricity usage enthalpy output. The global warming index, <∆T-biosphere> since 1950, presently +0.875 K, first became non-zero in the early 1970’s around the same time as natural gas usage began and has increased linearly by 0.0175 K/year ever since. Le Chatelier’s principle, applied to the dissipation processes of the biosphere’s ΔH-contours and [CO_2_] concentrations, helps to explain the global warming statistics.

## 1. Introduction

The global warming map over a 70-year period from 1950 to 2019 [1] is shown in Figure 1. This is the experimental data that any scientific theory, e.g., the greenhouse-gas hypothesis, must account for to become established scientific truth. To quote distinguished physicist Prof. Richard Feynman [2] *“It doesn’t matter how beautiful your theory is, it doesn’t matter how smart you are. If it disagrees with experiment, its wrong!”*

The Earth’s atmosphere concentric and time-span average temperature values <T> (z, ∆t)), and its gradients, known as the lapse rate = d<T(z)>/dz, have been assumed in climate models to be determined by a closed, reversible, adiabatic, expansion process against the constant gravitational force (g). The adiabatic lapse-rate hypothesis [3,4,5] predicts a constant negative temperature gradient which is a physical constant of air g/C_p_ (−9.8 K/km depending only upon air’s heat capacity (C_p_), as g is the earth’s gravitational constant. We have shown [3] that this idealized hypothetical process cannot be the causal explanation of the experimentally observed mean lapse rate (approx.−6.5 K/km) in the troposphere because it violates the 2nd law of thermodynamics.

The perturbing effect of transducer (‘greenhouse’) gases, that convert radiation to enthalpy, on a steady-state model atmosphere in a state of radiation balance equilibrium is that increasing the concentration of a weak transducer, CO_2_, could only have a net cooling effect, if any, on the concentric global average <T> (z = 0) at sea level and in the lower troposphere (z < 2 km) that determine climates. The global (time average) surface temperatures have been calculated by measuring temperatures at the surface of the oceans, and land temperatures at stations at the surface air temperatures. The data from 1949 to 2019 is accurate to within ~0.05 K. The results show that the global annual average increase is ~0.9 K for the 70 years since 1950 [1] This begs the question, “what is causing the global warming measured temperatures at station points on the biosphere surfaces around the globe and mapped in Figure 1”.

To find the answer to this question, we need to examine the experimental observations more geographically, and with some circumspection. The obvious features can be summarised as follows.

(i)The global warming effect is predominantly in the Northern hemisphere, and it is patchy with higher increases (>1.5 K) localised over land areas, though not necessarily densely populated;(ii)For most of the ocean surfaces the increase is between 0 and 1 K, averaging around 0.5 K;(iii)The global warming map hot spots (>2 K) appear in patches over land areas. The hot spots are (from west to east), Central America, Texas and the Caribbean land areas, Central Brazil, South Africa and Madagascar, Central Europe (Poland), Southern China, the Philippines, and Central Australia;(iv)There are some areas (light blue) where the average temperature has decreased slightly (less than 0.5 K); the most noticeable being a very large central intensive agricultural area of N. America, the great plains of the USA and Canada. Furthermore, there is a large global cooling region of the North Atlantic Ocean;(v)By far the greatest area of increase of 3+ K is an area all around the Arctic Ocean coastline regions from Alaska and Northern Canada, and from Russia north of Moscow half-way around the world to Siberia;(vi)There is a small 3 K+ hot spot on the edge of Antarctica.

These observations of spatial variations in the global increases in average temperatures at the biosphere land and sea surfaces are inconsistent with the CO_2_ greenhouse-gas hypothesis. The increase in concentration [CO_2_] associated with fossil fuel combustion over the 70 years will, by 2019, have uniformly dispersed by the forces of its chemical potential dissipation of all [CO_2_] concentration gradients around the globe by atmospheric motions, on time scales of days and weeks rather than decades, within the troposphere. To understand the global warming map in Figure 1, we must first consider the heat transfer processes, not only in the Earth’s atmosphere, but also in the Earth’s lithosphere. Whilst radiation balance transfer processes determine the average temperatures in the atmosphere [3,4,5], the temperatures and lapse rates in the Earths’ lithosphere are determined by geothermal processes [6,7]. Both thermal sources, *inter alia*, combine to determine the average temperatures at the interfacial biosphere. The surface heat transfer rates, along with the atmospheric radiation balances, determine the troposphere mean temperatures [8] and the Earth’s climates [9].

## 2. Earth’s Lapse Rates

### 2.1. Troposphere

The global time-average mean temperature at the surface of the Earth’s oceans and land areas, is around 285 K (12 °C). This is principally determined by a radiation balance. It can be demonstrated by simple one-dimensional models that if it wasn’t for the effect of water in the atmosphere, the temperature of the Earth’s surface would, on average, be around 50 K higher than it is. [3,4,5], and the biosphere and life on Earth would not exist as we know it.

Following decades of increasing the CO_2_ concentration, [CO_2_], the dispersion of the gas throughout the atmosphere could not account for these variations; we must conclude that the evidence presented in Figure 1 does not support the anthropogenic greenhouse gas, the increase in [CO_2_] and the global warming hypothesis. Rather, the evidence suggests that the more extreme increases in temperatures are of geothermal origin with a regional scatter of hotspots over land areas.

The graph of Figure 2 shows that at low-land ground level both the mean temperature <T> and the gradient of temperature (lapse rate) d<T>/dz of the troposphere coincide with these values in the earth’s lithosphere up to the tropopause at 10 km around the height of the highest mountains. This steady-state equilibrium can be disturbed in the areas of seismic or volcanic activity within the lithosphere.

### 2.2. Lithosphere

Planet Earth has been slowly heating up within by radioactive processes and cooling down at the surface since its formation around 4500 million years ago. The present consequence of these very slow processes is that there exists within the Earth’s core a temperature gradient from 7000 K at the centre to around 290 K in the biosphere and adjacent atmosphere. The estimated temperature profile compiled from various sources is shown in Figure 3a. The average temperature of the atmosphere at sea level is 289 K: the average gradient for the lithosphere is ~5 K/km; the Earth’s atmosphere extends to around 100 km. The steep decline in the lithosphere lapse rate (d<T>/dz) where <T> is a concentric average, can be explained by the change in the heat transfer process being predominantly convection in the Earth’s mantles, which are fluid, to mainly conduction through the lithosphere, i.e., the Earth’s crust, which is solid. In regions of seismic activity, whence there are convective heat transfer processes with higher rates of heat transfer, the lapse rate is lower and surface temperatures higher, as a consequence.

Figure 3a shows that there is a steep increase in the Earth’s radial lapse rate, i.e., averaged along the radii up to depth z. This increase is caused by the increased resistance to the heat transfer processes when the heat transfer mechanism changes from being predominantly convection in the fluid upper mantle, to predominantly conduction in the solid lithosphere. Not all the heat that reaches the land and ocean surfaces from the cooling gradient is transferred homogeneously around the Earth’s biosphere, as it is a continual heat transfer process that is permanently changing the overall energy budget balance at the biosphere where the temperatures in the global warming map have been measured.

Of special significance is the continuity of the lithosphere lapse rate (~5 k/km) that extends to the highest mountains, such as the Himalayan peaks roughly 10 km high with temperatures around–40 K of the same order as the temperatures within the troposphere at those heights above sea level. At sea and low land levels, both the mean temperatures and mean lapse rates of the lithosphere and troposphere coincide with the biosphere averages.

### 2.3. Biosphere

The biosphere can be defined as the interface between the troposphere and the lithosphere. It circumscribes the entire globe, varies in width from sea level to the depths of the oceans (~10 km) and to the height of the highest mountains at the atmospheric tropopause (~10 km). It is in a permanent state of heat flux. The global average temperature that is used to define global warming and the data in Figure 1 is measured at the biosphere surface air temperatures of the land areas to the highest mountains, and ocean surfaces at sea level.

Figure 4 shows that in the regions where the lithosphere is stable, closed and solid, the heat conduction from the lithosphere to the biosphere is relatively low, less than 40 mW/m^2^, but it can increase 10-fold in regions of seismic and volcanic activity around the tectonic plates beneath the oceans to heat transfer rates of 350 mW/m^2^. These figures are small compared with the solar energy arriving on half of the Earth’s surface every day: the net mean solar radiation energy is around 240 W/m^2^, i.e., of the order 1000 times greater. The heat transfer rates in Figure 4, however, are of the same order as the temporal and regional fluctuations in the global energy balance at the biosphere surface temperature monitoring stations.

Prior to the first recorded global warming effect, the annual average temperatures at all the recording stations around the world remained constant with <∆T>(∆t), the time-average change in the mean temperatures, in one decade, for example, exactly zero. Therefore, the net heat flux, on average (considering of all the sources of geothermal and sunlight heating, as well as day and night black-body radiation cooling, and transducer gas cooling and light scattering effects), [3], on balance, and, must by definition, also total exactly zero. A local heat transfer increase in any small region of the land surface, therefore, will cause a slight decrease in the local lithosphere lapse rate and an associated geothermal heating effect where temperatures are recorded in air at the land surface. In the following section we examine the evidence that fracking operations can indeed change the heat transfer rates, and the effect on average temperatures is not negligible. In Section 5, we will obtain the quantitative estimates to show that an increase in geothermal heat transfer rates in the lithosphere-biosphere interface will cause an increase in energy, enthalpy and entropy, and hence also cause a slight rise in average temperatures at the recording stations of the biosphere atmosphere interface.

The experimental data in Figure 1 showing the global warming hotspots, when viewed alongside the known intensive exploration and recovery shale gas regions, is the compelling evidence for fracking as the main cause of regional global warming above 1 K. We can obtain further evidence by quantifying an estimated increased heat transfer rate from Figure 4, i.e., between 40 and 350 mW/m^2^. Figure 4 shows that the mean lapse rate at the surfaces of the lithosphere is on average greater over the solid regions with widely differing thermal conductivities, than in the land surface regions contacting the biosphere. By turning the top 10% of the lithosphere up to 40 km in shale gas basins where hydraulic fracturing has been in operation, the practice has created seismic activity. In turn, this enhances a higher net thermal conductivity in the fractured network of the drilling operations by geothermal convection processes. From the heat transfer data in Figure 4, we estimate that an increase in the geothermal heat transfer of 100 mW/m^2^ could account for an increase in the mean local average temperature by 3+ K, and hence explain the global warming hotspots seen in Figure 1 [7].

## 3. Hydraulic Fracturing

### 3.1. Fracking and Thermal Conductivity

Hydraulic fracturing (‘fracking’: Figure 5)) involves forcing a liquid under high pressure from a wellbore against a rock formation until it fractures. The injected fluid contains a proppant—small, solid particles, usually sand or a man-made granular solid of similar size, that is forced to wedge open the expanding fractures. The proppant keeps the fracture open, allowing hydrocarbons such as crude oil and natural gas to flow more easily from the additional surface area to the rock formation provided by the fractures back to the wellbore (the drilled hole) and then to the surface. This process can extend to depths and lengths of up to 20 km and can also cause further seismic activity that will increase the thermal conductivity of the earth’s lithosphere in and around the geographic regions that have been affected by these operations.

The main reason for the steep decrease in average temperature in the lithosphere (Figure 3a) is the change in thermal conductivity mechanism from convection in the upper mantle to the much slower conduction in the solid lithosphere (Figure 3b). We note at this stage that every single fracking operation opens fissures that are eventually refilled with wastewater and drilling fluid mixtures that create a network of convective heat transfer pathways that will increase the heat enthalpy input to the atmosphere at the solid surface of the biosphere. At present, there are more than 2 million known recorded such fracking footprints on the Earth’s land surfaces plus countless unregistered fracking operations worldwide.

**Figure 5 entropy-24-01316-f005:**
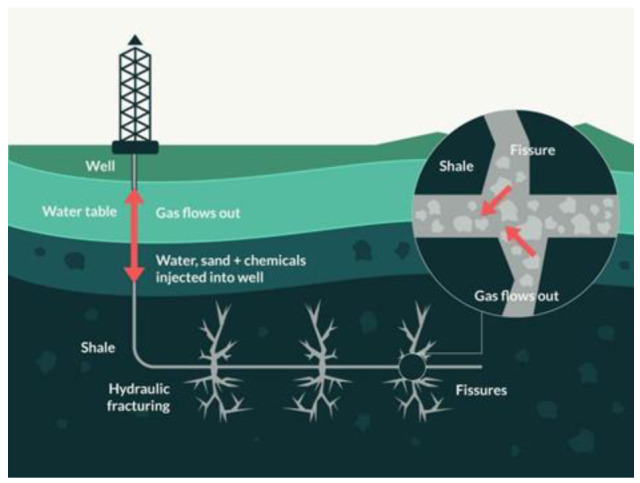
During fracking, a mixture of water, sand and chemicals–known as fracking fluid–is pumped under high pressure into a well to fracture rock and release hydrocarbons. These hydrocarbons are extracted at the wellhead together with wastewater that contains a mixture of fracking fluid and formation water. The wastewater can be disposed of by injecting it underground through deep wells. The well depths and fracking length range can vary from 100 m to 20 km [10].

### 3.2. USA Fracking Statistics

To estimate the worldwide problem, we examine the statistics from the United States [11] as being the most reliable and accessible regarding the rapid increase in the extent of hydraulic fracture operations (Figure 6) that lead to increases in heat flux in those areas of the lithosphere. The USA is a good example also because the hydraulic fracturing operations cover vast areas that include a high population, Texas, for example, and Alaska with a very low population, both shown as hotspots on the global warming map in Figure 1.

The hydraulic fracture process involves drilling a well vertically to a certain depth and then bending the path of the drilling until it extends horizontally. Because they are longer, and the drilling process is more complex, a horizontal well with a hydraulic fracture process produces more crude oil and/or natural gas. This method also results in horizontal wells having more drilled footage than vertical wells—hence the total footage drilled using horizontal drilling techniques surpassed vertical footage before the actual number of horizontal wells surpassed the number of vertical wells. In 2016, the total drilled footage in the USA exceeded 4 million metres, about 3 million of which were hydraulically fractured and horizontally drilled. The length of the horizontal section, or lateral, can range from hundreds of metres to tens of kilometres.

### 3.3. Fracking and Earthquakes

Both extraction and the underground injection of fluids have been shown to cause earthquakes. The scale of these events varies widely, ranging from relatively small earthquakes, such as the magnitude 2.3 in Blackpool UK [12], to the 5.7 magnitude earthquake in Prague, Oklahoma in 2011 [13]. There have been at least two seismic events of concern with magnitudes equal or larger than 7.0, both in Uzbekistan where shale-gas is recovered by fracking operations. It is not known whether these major earthquake events are coincidental with fracking or not.

The factors that are responsible for the fracking induced or triggered seismic events need more research as there is overwhelming evidence for an association between both, based on observations made concerning fracking in the vicinity or in the presence and orientation of tectonic faults, near locations of stress in the fault subsurface, and at the depth of and relation between the faults and the whole region around fracking process. Many fracking operations now extend to depths and radii of 20 km. The shale company Cuadrilla had been fracking at a site near Blackpool, UK when the earthquake with a magnitude of 2.9 was recorded near the UK’s only active shale gas site. It was the third recorded earthquake in the vicinity of the fracking operation in less than a week. The tremor was stronger than those that forced Cuadrilla to suspend UK operations in 2011.

Wastewater disposal wells typically operate for longer durations and inject much more fluid than is injected during the hydraulic fracturing process, making them more likely to induce earthquakes. In Oklahoma, which has the most induced earthquakes in the United States, 2% of earthquakes can be linked to hydraulic fracturing operations. Given the high rate of seismicity in Oklahoma, this means that there are still many earthquakes induced by hydraulic fracturing. The remaining earthquakes are induced by wastewater disposal. The largest earthquake known to have been induced by hydraulic fracturing in the United States was a magnitude 4.0 earthquake that occurred in 2018, in Texas [13].

The evidence for enhanced seismic activity connected to fracking and the shale gas industry is now overwhelming and irrefutable [14]. This effect on increasing the heat transfer processes in the lithosphere is entirely to be expected. At this stage, however, we do not have a detailed mechanism, and it is difficult to quantify the effects on the regional increases in the rates of heat transfer at the lithosphere surface layers down to 50 km in some instances.

### 3.4. Explosive Pressures

The hydraulic fracture process itself *de facto* works by inducing minor earthquakes. The drilling fluids used are basically colloidal dispersions of fine powdered silica or sand, in water with other solutes that that are rheology modifiers. Hydraulic fracture drilling fluids are Bingham plastic solids with a low yield stress. The plastic flow occurs when a shear yield stress is imposed, e.g., by a drill, and is highly thixotropic, becoming increasingly less viscous the greater the shear rate.

If the properties of water in the hydraulic drilling fluid are unaffected by the presence of the dispersed sand and rheology modifiers, it will begin to turn into steam with around 1000-fold increase in pressure in a confined space, when it encounters surfaces in the lithosphere with temperatures exceeding around 100 °C (373 K), the boiling point of water. We can see from Figure 3a above that the average temperature in the lithosphere increases with the depth by 5 K/km. With operating depths of up to 20 km, the fracking fluid can create enormous pressure waves that one normally associates with dynamite-like explosives, with far reaching seismic effects, besides opening channels for the flow of gas or oil from much further afield.

## 4. Fracking Hotspot Geography

### 4.1. Arctic Circle: Alaska, Canada Russia

Data from all of the various sources show the companies and the amounts of reserves and production. More shale gas from fracking operations is now produced from the Arctic regions of Alaska, Canada and Russia, (Figure 7) than the rest of the world combined (Figure 8, Figure 9, Figure 10, Figure 11, Figure 12, Figure 13 and Figure 14). This explains the widespread dark brown 3+ K ring of geothermal global warming all around the Arctic Circle, as seen in Figure 1. Russia is the world’s largest producer, and Siberia has the leading hotspot 5+ K.

### 4.2. Texas, Central America and the Gulf of Mexico

Hydraulic fracturing technology was developed and first used in Texas, USA, in the 1950s, but the state does not keep track of wells that use the hydraulic fracture practice, so the numbers may be unreliable (Figure 8). In 2021, the Texas state government halted the issuing of hydraulic fracturing permits after the technology was linked to an increase in earthquakes.

### 4.3. Brazil

According to Brazil’s federal energy research company (EPE), it has 14 sedimentary basins covering 2.34 Mkm^2^ with potential non-conventional resources. Together, the Parecis, Parnaíba, Reconavo and São Francisco basins are estimated to hold a natural gas volume of around 15.1 Tm^3^ (trillion cubic meters), making Brazil the 10th biggest shale gas resource globally. The largest output of 3511 billion m^3^ in the west central area (Figure 9: yellow circle) coincides with the central dark brown (3+ K) hotspot seen in Figure 1.

**Figure 9 entropy-24-01316-f009:**
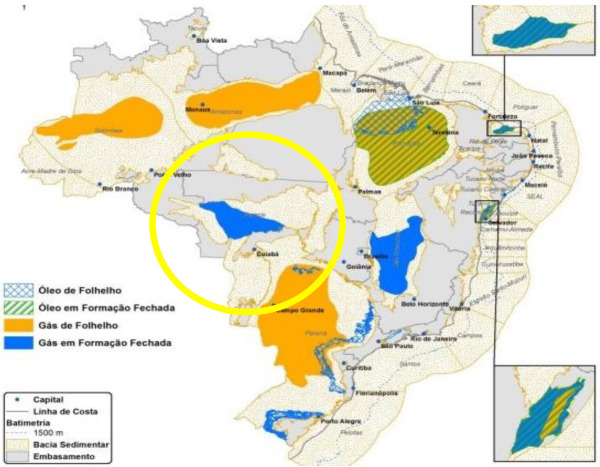
Brazil shale gas resources: the large dark brown 3+ K global warming hotspot, as seen in Figure 1, coincides with the most productive of the fracking operational facilities in the yellow circle [Appendix A (8)].

### 4.4. South Africa and Madagascar

The darkest brown hotspots seen in Figure 1 are seen (Figure 10) to be centred in the areas of greatest activity, namely the west coast countries of Angola and Namibia, and also Mozambique and Madagascar (not shown on map). The yellow circle 2+ K hotspot spans the whole island, as seen in the global warming map Figure 1. 

**Figure 10 entropy-24-01316-f010:**
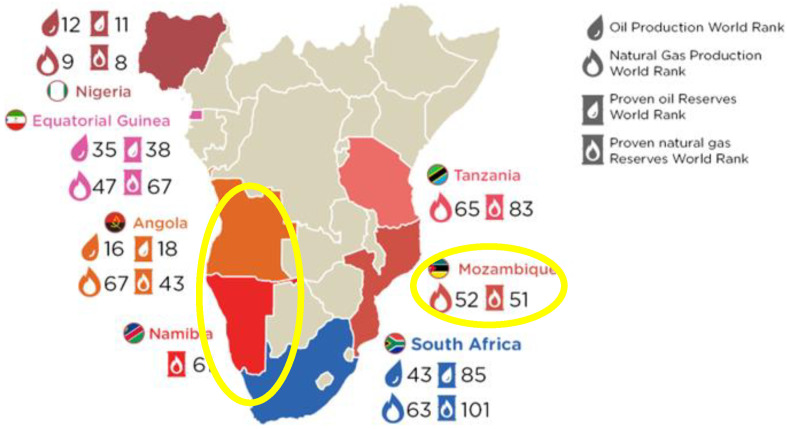
Offshore exploration developments in Southern Africa: the symbols are: (upper) world ranking of the oil production and natural gas production: (lower) proven oil reserves and proven natural gas reserves [Appendix A (9 and 10)].

### 4.5. Europe

The largest area of exploration and recovery activity by fracking operations in Europe is centred in Poland (Figure 11 yellow circle), Europe’s largest natural gas and fracking operator. The neighbouring countries of Romania and Bulgaria have presently suspended fracking operations for environmental reasons. This ultra-hotspot (3+ K) area is in the same geographic location as the dark brown 3+ K global warming hotspot seen in Figure 1 and Poland is shown with a yellow circle. The central hot spot fracking area of the UK has registered several earthquakes that have been associated with on-going fracking.

**Figure 11 entropy-24-01316-f011:**
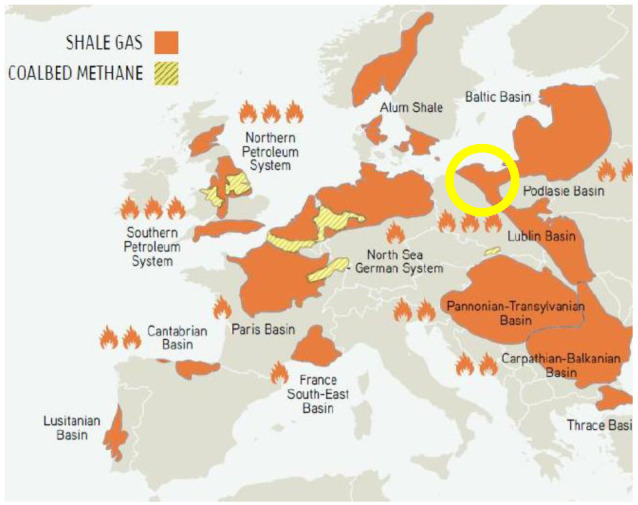
Natural gas resources in Europe [Appendix A (11)] the hotspot 3+ K in Figure 1 is Poland (yellow circle).

### 4.6. China and the Philippines

The Manilla region, as a global warming hotspot, shows no gas production areas on Luzon Island in the Philippines (Figure 12). Since the 1990’s, however, the Philippine government has increasingly used fracking technology to extract heat from the lithosphere hot rocks to turn water into steam, and steam into electricity. Geothermal energy by fracking now accounts for 20% of the country’s electricity supply. Figure 1 showing a 2+ K hotspot suggests that this may not be a “sustainable” energy production.

**Figure 12 entropy-24-01316-f012:**
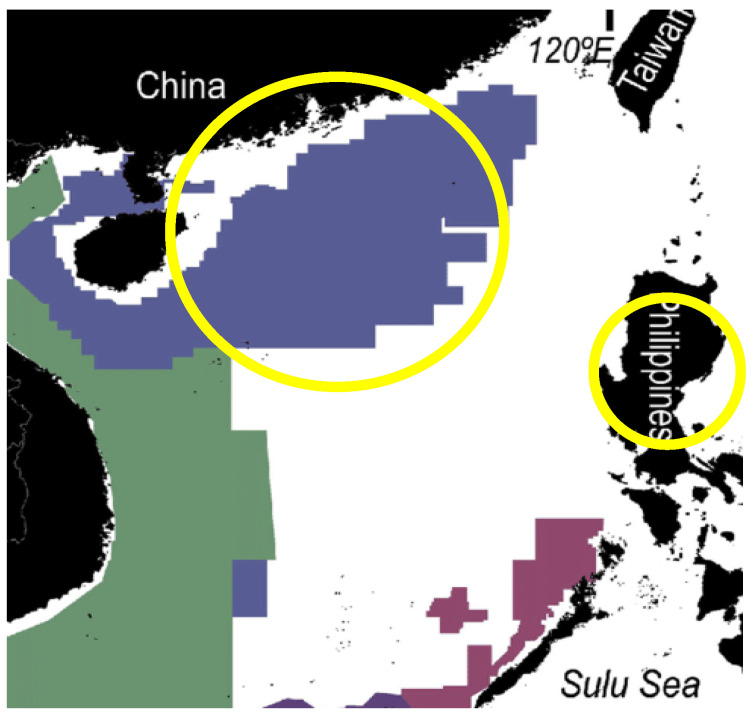
Allocated areas of shale gas reserves in Southern China and the Philippines. The blue area is the Chinese fracking operational region, and the purple areas are the Philippines. [Appendix A (12 and 13)].

### 4.7. Australia

Gas is Australia’s third largest energy resource after coal and uranium. Conventional gas resources are widespread both on and offshore, occurring in 14 different basins, but most of the resources are off the north-west margin in the Bonaparte, Browse and Carnarvon basins. The fracking area extends from the west coast to the east coast, as shown on the map. Australia has more than 40,000 coal-seam gas drilling sites that use fracking to crack the seams. Natural oils, gases and geothermal energy via steam and electricity are extracted. The yellow rectangle coincides with the 2 + K hotspot in Figure 1.

**Figure 13 entropy-24-01316-f013:**
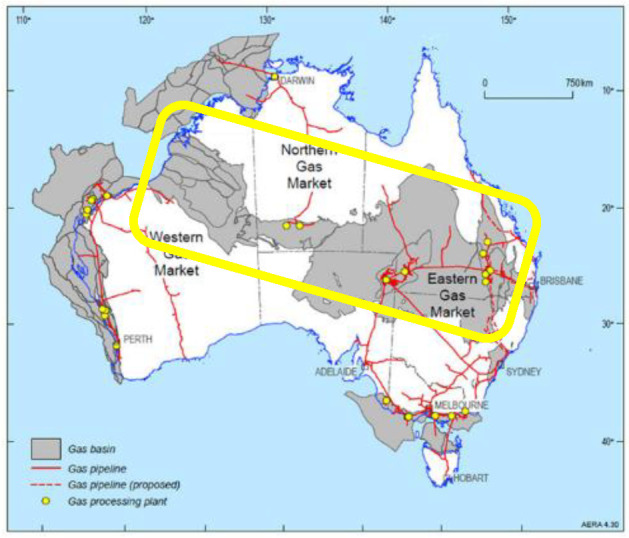
Australia shale-gas facilities: [Appendix A (14–16)].

### 4.8. Antarctica

Over the past 50 years, the west coast of the Antarctic Peninsula has been one of the most rapidly warming parts of the planet. This effect is not only restricted to the land but can also be noted in the Southern Ocean. The upper ocean temperatures to the west of the Antarctic Peninsula have increased over 1 °C since 1955. It has recently been established that the Antarctic Circumpolar Current is warming more rapidly than the global oceans.

**Figure 14 entropy-24-01316-f014:**
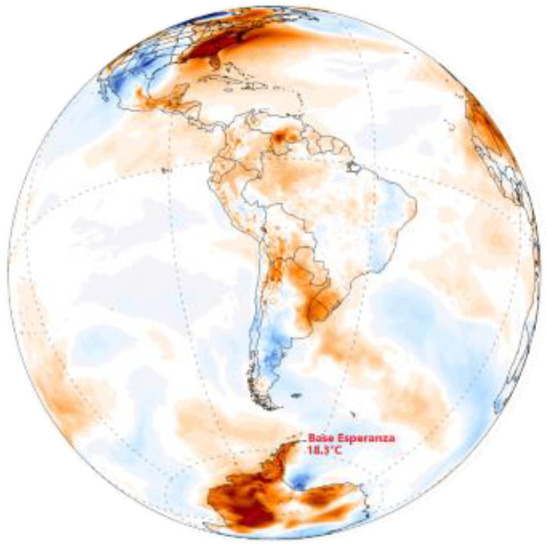
Global warming map for the last decade and showing the hotspot location in Antarctica. [Appendix A (17–19)].

## 5. Enthalpy ‘Footprints’

### 5.1. Thermodynamic Definitions

Heat and work from all fuels, including nuclear, as well as electricity usage from all sources, contribute positively to global warming. To understand and quantify the effect, if that is at all possible, considering the complexity of the systems, we must resort to the basic principles of the laws of thermodynamics. All thermodynamic subsystems (solid, liquid or gas, land, sea and air,), when they are at local or metastable equilibrium with the ambient temperature (T) and pressure (p), have an associated total amount of energy. This internal total energy of the system can only change by transferring heat (Q/J/kg) from one p-T state to another, or between states. A change in energy at constant pressure, such as for instance, one atmosphere, defines the state function enthalpy (H).

The temperature differences <∆T> depicted in Figure 1 are recorded from the air temperature at ground and sea levels around the globe. Consider an imaginary volume of air at one of these stations, for example, 1 g-mole, and this volume of air can expand and contract via an imaginary boundary that permits the passage of heat and light, so that it can equilibrate with the surroundings at an ambient temperature and pressure, that is changing periodically on time scales of days and months (seasons) and fluctuating randomly on all time scales beyond just a few minutes.

Every homogeneous fluid material has associated with it a heat capacity (C_p_) which is the heat (Q J/mol) to increase the temperature 1 K. The whole atmosphere obeys the ideal gas equation for diatomic molecules (C_p_ = 7/2 R where R is the gas constant per mole). To reach a good approximation, the heat capacity of air throughout the atmosphere, irrespective of the temperature or pressure, is 29.07 J/mol.K.

Given the definition of heat capacity C_p_, the laws of thermodynamics define enthalpy and entropy [15].

For infinitesimal changes (denoted by δ), since C_p_ is, in general, also a function of state.
1st Law: enthalpy is a state function: δH = δQ = C_p_ δT(1)
2nd Law: entropy is a state function: δS = δQ/T = C_p_ δlog_e_T(2)

These two laws tell us that the burning of all fuels, including non-fossil fuels such as nuclear fuel, all of which transfer an amount of heat Q, i.e., enthalpy and entropy (=Q/T) into the atmosphere, increase the enthalpy, the entropy, and the temperature (T) of the atmosphere, to some degree. All fossil fuels that burn to CO_2_ and H_2_O, will increase the enthalpy of the entire atmosphere by an amount equal to the total sum of all the enthalpies of combustion of all of the fuels consumed, integrated over time. If the atmosphere is initially at temperature T_1_, the temperature of the atmosphere will increase to T_2_ = T_1_ + ∆H/C_p_. Likewise, the second law tells us that if the heat energy transferred to the atmosphere is at temperature T_1_ of the original station sample, the entropy will have also increased by ∆S = C_p_ log_e_ (T_2_/T1). There are no means of recovering any of the energy ∆H so that it can be reused. The entropy footprint is defined as the available energy of the consumed fuel that cannot be recovered as it mainly exists at the ambient temperature of the atmosphere. In the case of burned fossil fuels, the entropy increase tells us that all the energy transferred, T∆S, is forever spent.

### 5.2. Steady State (Zero Global Warming)

The Earth’s climate has been changing for 4500 million years since its formation from condensing space gases in the sun’s orbit. Let us first consider the period 950–1950 AD when presumably there was no “climate change” on our timescale of recent natural history. Figure 2 and Figure 3 of the lithosphere and atmosphere lapse rates show that the Earth’s gradients of temperature are a time-independent constant so there can be no “thermodynamic equilibrium”; there is a permanent state of energy flux, on average, between earth and space. However, if there are no recorded average temperature changes, both the Earth and its atmosphere have equilibrated at a steady state with the net retention of the sun’s energy since the transducer molecules radiate some absorbed energy in all directions [16]. The net sun heating reception at the surface is approximately, 240 W/m^2^, balancing to a large extent the heat loss by the Earth’s black-body radiation [17]. The energy balance from all sources, albeit immensely complicated in origin, can be summarised in a simple equation.
<∆H> = + <∆H_S_> − <∆H_E_> + <∆H_G_> ± all others = 0total sunlight Earth IR geothermal(3) At a steady state, the details of the origin of the terms, their fluctuations day and night, months, seasons, etc., is irrelevant, if the global warming average <∆T> everywhere is zero. There are many major and minor terms of complex origin in the energy balance, but the fact that we can’t quantify all of these doesn’t matter. For a steady state, the total is exactly zero. The annual enthalpy budget at all of the temperature monitoring stations around the globe must balance for a steady state of zero global warming.

### 5.3. Fossil Fuel Enthalpy

When fossil fuel is burned, e.g., a hydrocarbon, such as gasoline (octane), for every 12.5 moles of oxygen is burned up, it is replaced by 16 moles of CO_2_ + 18 moles of H_2_O in the atmosphere. What would be the effect of this CO_2_, H_2_O and enthalpy of combustion footprints on <T> (z), we might ask, if all of the exhausts from all fossil fuels are the sum of all uniformly distributed in the atmosphere? Consider as an example, the balanced chemical equation for the combustion of liquid octane:C_8_H_18_ + 25/2 O_2_ → 8 CO_2_ + 9 H_2_O + 5430 kJ (=∆H)octane (l) oxygen (g) carbon dioxide (g) water vapor heat enthalpy(4) The amount of 116 g of liquid octane will generate 352 g of CO_2_, and 162 g of H_2_O plus a massive liberation of more than 5 MJ of heat enthalpy into the atmosphere. Let us assume that all the heat produced by all the fuels in the last 100 years has been emitted into the atmosphere. Of course, there are many other balancing processes during that period but, to begin with, just to assess the order-of-magnitude of the enthalpy output and assess whether it can create global warming data on the scale seen in Figure 1. We can do a simple calculation of what the increase in the atmosphere’s temperature would be if all this enthalpy were emitted at once and equilibrated throughout the atmosphere.

Figure 15 shows that the global energy supply systems have transformed dramatically since the industrial revolution of mid-nineteenth century. The global energy supply chart reproduced here details the global energy consumption from 1800 onwards. It is based on the historical estimates of primary energy consumption from Vaclav Smil, combined with updated figures from BP’s *Statistical Review of World Energy* [18]. An estimated energy consumption for the whole world from all sources is obtained by integrating an area over a 70-year period from 1949 to 2019, as shown to a good approximation by the area of the triangle in Figure 15. The integral ∆H = 8 × 10^6^ TWh (=2.88 × 10^22^ J): given that the heat capacity (C_p_) of the whole atmosphere is 5.6 × 10^21^ J/K, the ratio ∆H/C_p_ in a hypothetical thermodynamically reversible process, corresponds to a temperature increase of ~+5 K for the whole atmosphere.

This simple calculation confirms that the enthalpy footprint is not negligible compared to other sources of fluctuations from the pre-global warming 1949 temperatures to 2019. The calculation, however, assumes that there are no mitigating factors to counteract the gradual increase in enthalpy over a period of 70 years. The global temperature increase will be much less than this ~5 K maximum. It is diminished because the Earth was previously in a dynamical steady state. A thermodynamic principle comes into play to reduce the effects of the enthalpy footprint, invariably in the direction of restoring the energy balance at a new steady-state equilibrium. Unfortunately, the various effects of Le Chatelier’s principle can be inextricably related, complex, and difficult to quantify. The evidence from Figure 1, however, of Le Chatelier’s effects is a regional global cooling effect shown as patches of lightest blue areas.

### 5.4. Le Chatelier’s Principle

The earth’s atmosphere, at the surface of land and sea, is a dynamical thermodynamic system fluctuating on all time scales, yet, in a steady state, if there is no global warming at any monitoring point average p, T states remain constant, hence also the state functions enthalpy and entropy. If that steady state is changed, for example by increasing heat from the lithosphere, or enthalpy from fuels, then a fundamental principle of classical thermodynamics intervenes to alleviate the effect of change to some extent.

Le Chatelier’s rule [19] states: “***if a dynamic equilibrium is disturbed by changing the conditions, the position of equilibrium shifts to counteract the change to reestablish an equilibrium***”.

Thus, we can use thermodynamics, at least to predict the direction of the new equilibrium steady state, although it may be difficult to quantify accurately due to the diversity of counterbalancing sources. Taking into account all of the following cooling effects that continually reduce the enthalpy increase, from combustion of all the fuels used in 70 years, For the greater part of the biosphere, in regions where there are no increases by geothermal heating caused by fracking, the enthalpy footprint after all of these re-equilibration effects during the period of Figure 1, has reduced the global warming average enthalpy footprint to a small fraction of 1 K, as evidenced by the recorded temperature data.

The foremost counter effect of the increasing mean temperatures <T>(z) in the atmosphere by the emission of enthalpy of combustion in Equation (5) and increasing the enthalpy by the geothermal heat from fracking operations, will also have a counterbalancing effect in the heat loss from the Earth’s atmosphere by black-body radiation in all directions. If the temperature of the Earth’s surface increases by <∆T>, then the Earth’s surface will lose slightly more heat by black-body IR radiation by the ratio total atmosphere [T/(T + ∆T)]^4^ where T is in Kelvin. When T is 285 K, the change is the reduction in enthalpy ∆Hir in Equation (3) by the factor 0.986, for example.

Next, consider the effects of burning fossil fuels, using the example of octane, in Equation (4). This equation tells us that the three steady-state variables that determine the heat balance is to increase (i) the concentration of [H_2_O] water vapor in the atmosphere (ii) the concentration of [CO_2_] and (iii) the temperature, according to thermodynamic law equations (1 and 2) that have changed.

The steam footprint at first, seems negligible compared to the water already present in the atmosphere at the steady-state and therefore its equilibrium with the clouds, the oceans and the biosphere all remain undisturbed by gasoline emissions over several decades. Adding more water vapor to the atmosphere, albeit relatively small amounts, could produce a ‘steam footprint’ global cooling effect. This could happen because more water vapor leads to more extensive saturation in the troposphere, hence more cloud formation. Clouds reflect sunlight and reduce the amount of radiation heat that reaches the Earth’s surface to warm it. If the amount of solar warming decreases, then the temperature of the Earth’s surface and lower troposphere would decrease. In that case, the effect on average temperatures, if any, of adding more water vapor via the industrial ‘steam footprint’, that invariably accompanies the industrial ‘carbon footprint’, would cause global cooling, rather than warming. This will counterbalance the heating effect of ∆H to some unknown extent.

Similar considerations, however, apply to [CO_2_]. A corollary of such an argument, therefore, is that if the steam footprint is negligible, then the [CO_2_] concentration footprint from gasoline engines is probably also negligible. However, there is a far more significant footprint from doubling [CO_2_]. Photosynthesis is an endothermic chemical reaction catalysed by sunlight; Le Chatelier’s principle [19] tells us that heat, either by conduction or radiation, promotes the equilibrium in the direction of plant growth. Take the carbohydrate synthesis reaction, C_n_H_2n_O_n_ when n = 6 as an example.
6 CO_2_ + 6 H_2_O + [ћν] (sunlight) → C_6_H_12_ O_6_ + 6 O_2_ −1264 kJmol^−1^ (=∆H)(5) This exemplary reaction converts CO_2_ from air and water to carbohydrate, e.g., glucose (C_6_H_12_O_6_) and oxygen (O_2_) by using the radiation energy from the sun. Thus, we see that the first law of enthalpy balance in a cyclic process requires the solar energy term in Equation (3) to be reduced by all of the endothermic plant-growth photosynthetic processes on the Earth’s surface. This process, albeit difficult to quantify accurately, will counterbalance the global warming enthalpy heating footprint of all fossil fuels that also emit CO_2_ as Equation (4), for example.

For the atmospheric concentration [CO_2_], moreover, there is a balancing effect in the biosphere as Le Chatelier’s principle works towards the restoration of the chemical equilibria for the photosynthesis reaction that removes CO_2_ from the atmosphere for plant growth, replacing the oxygen. If the concentration [CO_2_] increases, all vegetation in the biosphere will grow more, and faster. So, the two effects of increasing {CO_2_} in the atmosphere are to decrease both [CO_2_] and to decrease <T> in the biosphere. This global cooling effect of photosynthesis could explain the very slight cooling, less than 0.5 K, in the great plains intensive agricultural farming areas of the USA and Canada seen as the light blue areas in Figure 1.

### 5.5. Henry’s Law and Photosynthesis

The biosphere global average temperatures are determined, to a large degree, by the thermodynamic properties of the oceans. Both sea and air surfaces, at sea level, represent 2/3 of the biosphere’s global average temperature dataset by location. CO_2_ is much more soluble in water than the other atmospheric gases. The thermodynamic equilibrium solubility of CO_2_ in water is given by Henry’s law constant of the proportionality ratio of the solubility and the atmospheric partial pressure of CO_2_. Taking just the dimensionless concentration ratio of {CO_2_}-oceans to [CO_2_]-atm., Henry’s constant is 0.83 at T = 285 K; the net result is that if the concentration [CO_2_] has doubled in the atmosphere, then so also has the concentration [CO_2_]-ocean in the steady-state thermodynamic equilibrium. Henry’s law constant increases as the temperature decreases, so the concentration [CO_2_] increases with the depth below sea level. The net result is that the CO_2_ oceanic cooling effect of the conversion of CO_2_ to seaweed and plankton by endothermic photosynthesis, producing an additional trillions of tonnes in the oceanic entirety, will also to a large extent, not yet quantified, counterbalance the enthalpy increase within the atmosphere by fossil fuel production of heat plus CO_2_.

### 5.6. Electricity Enthalpy

All electricity contributes directly to global warming, irrespective of its production source: coal, gas, oil, nuclear, hydropower, wind, solar, turbine or tidal, in order of capacity. Electricity increases the energy industry’s enthalpy footprint when it is used. Although the renewable sources of electricity are said to be “sustainable” there will be a very small enthalpy footprint price to pay when and where it is converted to heat or work. The global total may not be negligible and can be roughly calculated for all usage: lighting, heating, electric motors, charging batteries, etc., into wattage, by using Joule’s law for the conversion of electricity power into heat by the first law of thermodynamics conservation of energy. Joule’s heating law [15] of electric motors can be quantified as the enthalpy equivalent when electric power is spent converting it into work. The net electrical power enthalpy output can be estimated from the world usage since 1985, as shown graphically in Figure 16.

We calculate that the total heat generated from electricity in a 70-year period from 1950 to 2020 corresponds to approximately ∆H = 3 × 10^21^ Joules. Given the heat capacity of the whole atmosphere 5.6 × 10^21^ Joules/K this could increase <T> of the entire atmosphere by ∆H/C_p_, around 0.5 K. This is only 10% of the enthalpy output from fossil fuels since 1949 in Figure 14, but it is not negligible. We also note that, whereas the enthalpy output of fossil fuel burning directly is accompanied by the emission of transducer gases H_2_O and CO_2_, that have a net cooling effect of the biosphere, electricity has no such counterbalancing benefits.

**Figure 16 entropy-24-01316-f016:**
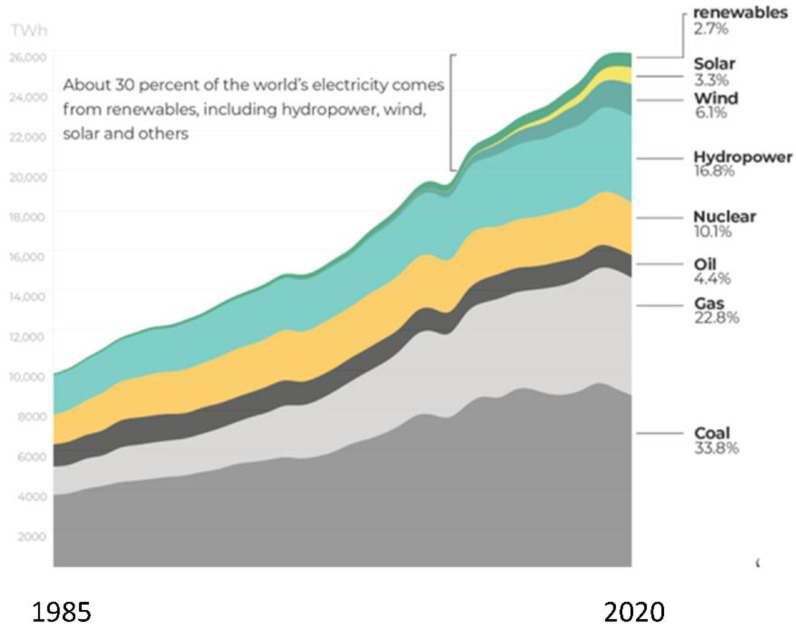
The world’s electricity production and consumption by source from 1985 to 2020 (linear scale): the total consumption interpolating back to 1950 and integrating, is estimated to be 26,000 TWh × 70 years/2 = 0.91 × 10^6^ TWh.

## 6. Conclusions

We have found that the global warming map, Figure 1, is consistent with hydraulic fracking operations as the root cause of the hotspot temperature increases in many countries. The fracking-geography maps, reproduced in Figure 7, Figure 8, Figure 9, Figure 10, Figure 11, Figure 12 and Figure 13, show precisely the same global warming hotspots. Namely, the Arctic Circle from Alaska to Siberia, Central Brazil, Poland in Central Europe, Southwest Africa and Madagascar, and the band across Central Australia and the Southeast EU countries, Southeast China, the Philippines, and a hotspot in Antarctica. That these hotspots do not disperse with time, to increase slightly the rest of the global warming effect, above the 2/3 Earth’s ocean surfaces. We can conclude that the global warming effects in these regions, are permanent changes to the overall steady-state energy balance at the biosphere-troposphere interface where the temperature increases regionalised in Figure 1 were recorded. The hotspot changes seen in Figure 1, that are centred on regions of the most extensive fracking operations, are permanent and practically irreversible.

In this analysis, we have also considered the fossil fuels of the energy industry and electricity, that have enthalpy emission footprints, and can explain worldwide global warming to an unknown small extent. We have not considered separately the contribution to global warming of the enthalpy emissions from alternative sources, all of which will also have an enthalpy output that will cause global warming to some very small extent, including, geothermal energy, with and without fracking and nuclear energy. We have observed that some countries, notably the Philippines and Australia (Figure 12 and Figure 13), are becoming increasingly reliant upon geothermal energy that is generated by fracking [20,21].

### 6.1. Constructal Law and Fracking

Whereas changes to the thermodynamic equilibria are predictable by Le Chatelier’s principle, it has been suggested that the heat transfer processes relating to global warming may be predictable by a concept that has been named the “constructal law” [22]. “Constructal” is an adjective that does not otherwise exist in the English language: is a more nebulous concept than Le Chatelier’s principle, although more widely applicable with more wide-ranging experimental support. Simply stated, heat and mass transfer processes that are subjected to any natural framework perturbation, will respond in the direction of an increase in transfer rates. Clausse et al. have recently applied the theorem to a very simple model of atmospheric convection processes [22]. We see here, however, that the concept may be more relevant to the effect of fracking operations, as a perturbation of the existing heat-transfer steady-state, that has only resulted in increased rates of heat transfer from the outer lithosphere to the biosphere.

### 6.2. Sustainability and “Net Carbon Zero”

Finally, what does this tell us about sustainability, i.e., zero global warming in a new steady state? These conclusions are consistent with the experimental results of Figure 1. That is also consistent with the global average temperature change (Figure 17) measured to a high accuracy (±0.01 K) as evidenced by the agreement to the periodic fluctuations of 2–4 years from five different scientific institutions with temperature monitoring stations. The global average increase for the biosphere land and sea surfaces, i.e., the global warming index presently stands at <∆T> 0.875 ± 0.005 K and is increasing linearly by 0.175 K every 10 years.

To address the sustainability issue, the linear increase 0.0175 K/year since 1970 in Figure 17 must be arrested for a return to zero global warming. “Net zero CO_2_” will exacerbate rather than reduce global warming. We conclude with the following observations regarding the measures necessary for the stabilisation of a global average temperature

(i) The 3+ increases (dark brown in Figure 1) have probably permanently increased the global average temperature in those regions. Unlike the combustion of fossil fuels, fracking increases the thermal conductivity of the lithosphere regionally and will increase the localised global warming index for countries that extensively exploit the technology. The warmest hotspot is an alarming 5+ K in Russia’s Northern Siberia region. Only a moratorium on fracking operations can arrest this global warming effect.

(ii) To allow the Earth to come to a new sustainable steady state for which the global warming definition of the energy budget balance is again exactly zero, the increase in the world combustion of all fuels, except for renewable biofuels, but including the usage of electricity, must be maintained at constant values, and not allowed to increase steeply with time as seen in Figure 15 and Figure 16;

(iii) Increasing the concentration of CO_2_ cannot be a causal explanation of global warming. It acts as a global coolant via photosynthetic processes. According to Le Chatelier’s principle, when the concentration [CO_2_] increases in the atmosphere, all photosynthetic vegetation growth process reactions progress faster and to a greater product extent. The biosphere is an open system of oceans and atmosphere also in chemical equilibrium, with no permanent mean gradients of chemical potential, i.e., of [CO_2_] concentration between oceans and atmosphere at steady state. Henry’s law ensures that the endothermic cooling effects of {CO_2_] extends to the Earth’s oceans. There is also a benefit of increased atmospheric [CO_2_] for the food and agricultural industries and biofuel production.

(iv) A corollary of (ii) and (iii) above, is that switching to electric battery driven cars for transport, for example, or from fossil fuels to nuclear, are false premises that will not reduce global warming. In fact, it is exactly the opposite. Putting the enthalpy into the atmosphere via electricity production and usage, without replenishing the CO_2_ will diminish the CO_2_ photosynthesis cooling effect, that can be seen in Figure 1 (light blue) for the intensive agricultural areas of the great plains of Central North America. By the same token, the only truly sustainable fuel that can deliver unlimited energy without global warming is biofuel. The importance of developing biofuel technology in the present context is emphasised and discussed in detail in a recent review article [22].

(v) Likewise, it does not make any sense to continue with worldwide increases in all fuel and electricity usage by simply replacing fossil fuel with nuclear generated electricity. That will not help to reduce global warming, without a reduction in the nuclear steam production and electricity enthalpy footprints. Nuclear fuel consumption and electricity usage do not contribute to the replenishment of atmospheric and oceanic CO_2_ concentrations that are necessary for biofuels, the food and agriculture industries, and as a global coolant via endothermic photosynthesis processes in the biosphere [23].

(vi) Methane (CH_4_) has also been suggested as a global warming contributor in the greenhouse-gas hypothesis. Methane concentrations [CH_4_] in the troposphere cannot significantly affect the biosphere global average temperatures. Notwithstanding some relatively minor fracking spillages, the distribution in the troposphere is effectively uniform and has increased from 0.00007% in 1780 to 0.00018% in 2020. It exists in the atmosphere as an ephemeral unstable reactant with respect to oxidation to CO_2_ + 2H_2_O (=CH_4_ + 2O_2_) with a CH_4_ half-life of around 10 years. This increased CO_2_ production is negligible compared to the direct output of CO_2_ from fossil fuels. Hence, we can conclude that neither CH_4_, nor CO_2_, concentrations in the troposphere contribute to the recorded global warming results reproduced here in Figure 1 and Figure 17;

(vii) The greenhouse gas hypothesis flagship of ‘supporting evidence’ is the correlation between CO_2_ concentrations and temperature rises of the last 50 years [24]. This correlation, however, is wholly consistent with the fossil fuel enthalpy footprint causing global warming, and the accompanying [CO_2_] footprint (e.g., Equation (4)) being an otherwise rather beneficial side effect for the biosphere and agricultural industries. Increasing [CO_2_] can to some degree alleviate global warming by Le Chatelier’s principle applied to photosynthesis in the biosphere. Replacing fossil fuel with nuclear fuel, for example, without stabilising the total enthalpy output to a constant value, will only help to maintain a linear global warming effect, as seen in Figure 16, as will also replacing fossil fuel powered cars with electric motors with the same net enthalpy output.

The increased concentration [CO_2_] plays a central important role as the global coolant in the biosphere, land surfaces and oceans, by Le Chatelier’s principle of thermodynamic processes, that counteract the enthalpy footprints. The greenhouse gas hypothesis has been blindly accepted as a scientific truth in contradiction with the established scientific method. Consequently, research into alternative explanations for global warming based only upon classical thermodynamics and experimental evidence have not been supported by national research funding foundations. This climate-change false premise has given rise to “carbon net-zero” objectives on an industrial worldwide scale. A blinkered pursuit of atmospheric [CO_2_] reduction as the only antidote to global warming is counterproductive. The global implementation of this policy will not arrest the temperature rise. As seen in Figure 16, and could lead to steeper increases in the biosphere’s average temperatures from now on.

## Figures and Tables

**Figure 1 entropy-24-01316-f001:**
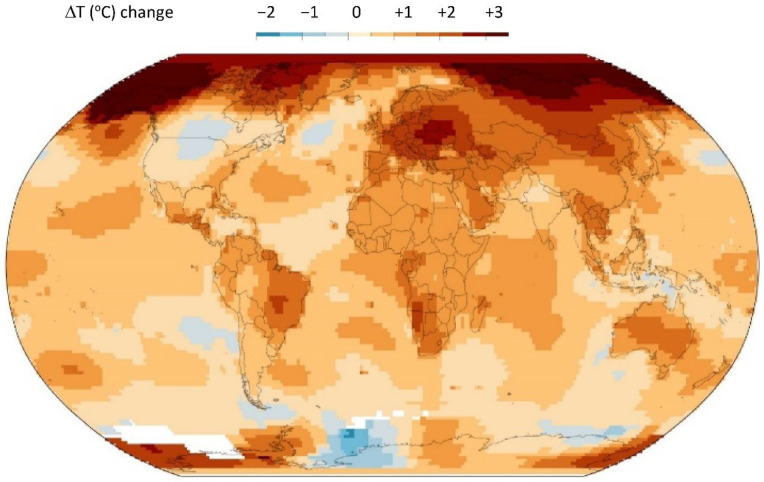
Global warming map of change in average temperature, from 1950 to 2019, at ground and sea levels: the color-coding changes are as shown above; the white areas near antarctica are “no data available”: New York Times 15 January 2020 [1].

**Figure 2 entropy-24-01316-f002:**
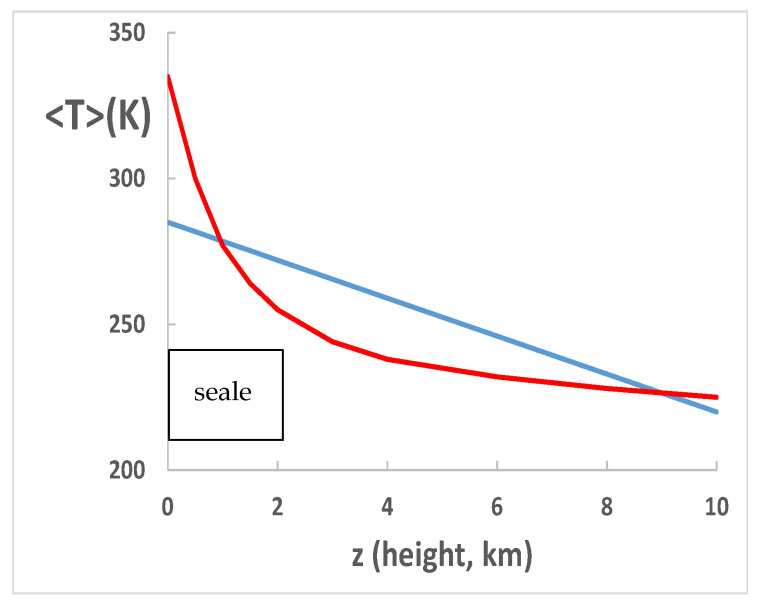
Troposphere average temperature (<T>) profiles: blue line is the experimental T-profile showing a constant lapse rate (gradient) of −6.5 K/km: red line is the computer model [5] effect of radiation balance without transducer gases H_2_O and CO_2_. The effect of the transducer gases, predominantly H_2_O, is to reduce the temperature at sea level by around 50 K from around 335 to 285 K, on average.

**Figure 3 entropy-24-01316-f003:**
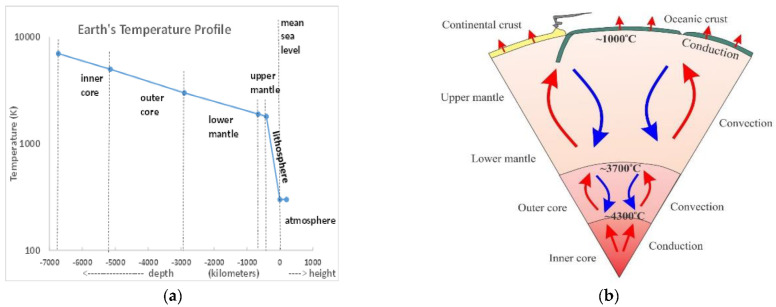
(**a**) An average temperature profile of the Earth and its atmosphere: (**b**) schematic picture of the heat transfer processes from the Earth’s inner core to the lithosphere.

**Figure 4 entropy-24-01316-f004:**
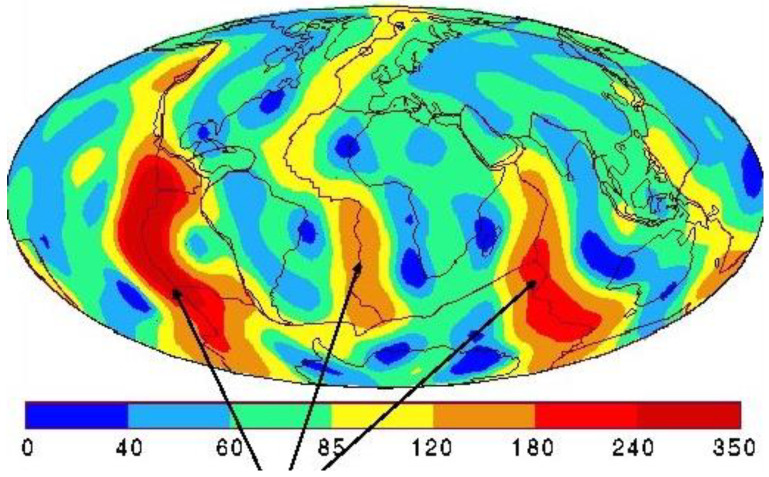
Earth’s surface geothermal heat flux from the lithosphere to the biosphere land and sea surfaces: colour coded heat transfer rates are in mW/m^2^: the arrows indicate the tectonic plate boundary spreading centres with heat transfer rates averaging 350 mW/m^2^ in those regions.

**Figure 6 entropy-24-01316-f006:**
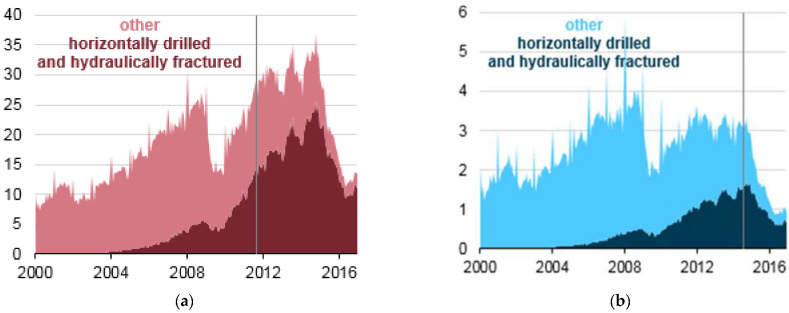
(**a**) Statistics from the USA shows millions of feet drilled per month for oil and gas, by type of drilling (2000–2016): fracking exceeded all other types in 2011, as marked, and since 2016, has now nearly reached 100% of all recovery operations (**b**) Statistics of the well-type drilled in thousands per month in September 2014. Most wells are horizontally drilled and hydraulically fractured [11].

**Figure 7 entropy-24-01316-f007:**
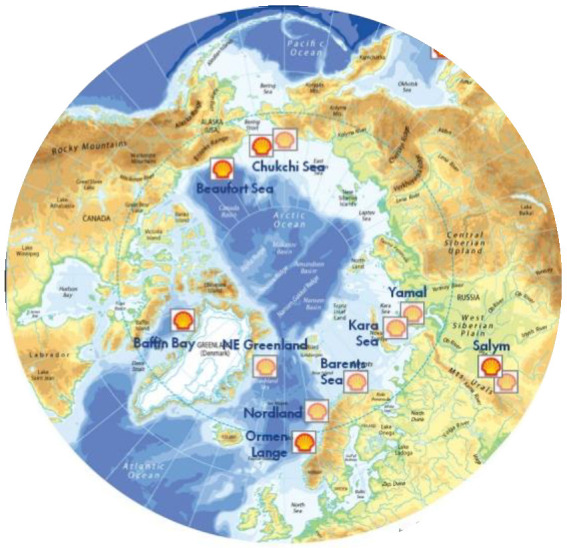
Map of the Arctic Circle showing all of the exploration and recovery drilling sites with the (official) major oil companies [Appendix A (1–6)].

**Figure 8 entropy-24-01316-f008:**
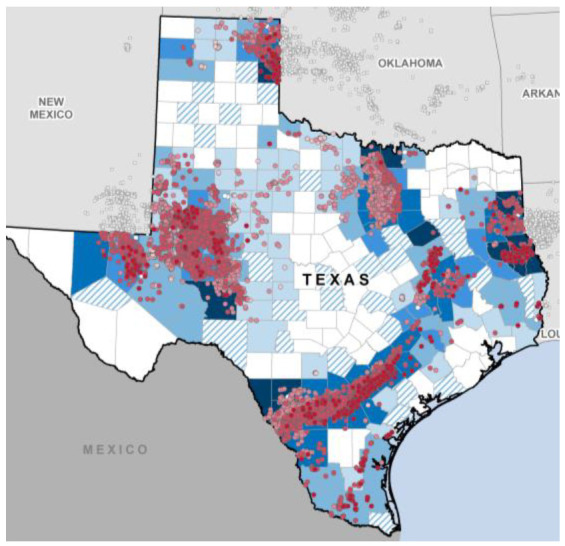
Fracking and oil-gas recovery map of Texas, USA: the exploration sites are shown as purple dots and are so numerous that they merge with some intensive oil-gas producing regions [Appendix A (7)].

**Figure 15 entropy-24-01316-f015:**
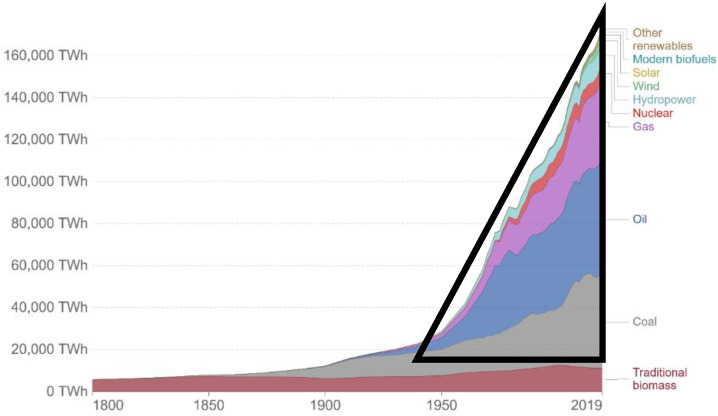
Global primary energy consumption by source since 1800. The primary energy is calculated based upon the “substitution method” that accounts for inefficiencies in fossil fuel production by converting non-fossil energy into the energy inputs if they had the same conversion losses as fossil fuels [18]. Notice that a significant steep increase in the combustion of natural gas (purple) begins in the early 1970’s, i.e., around the same time as the onset of the biosphere average non-zero global warming index.

**Figure 17 entropy-24-01316-f017:**
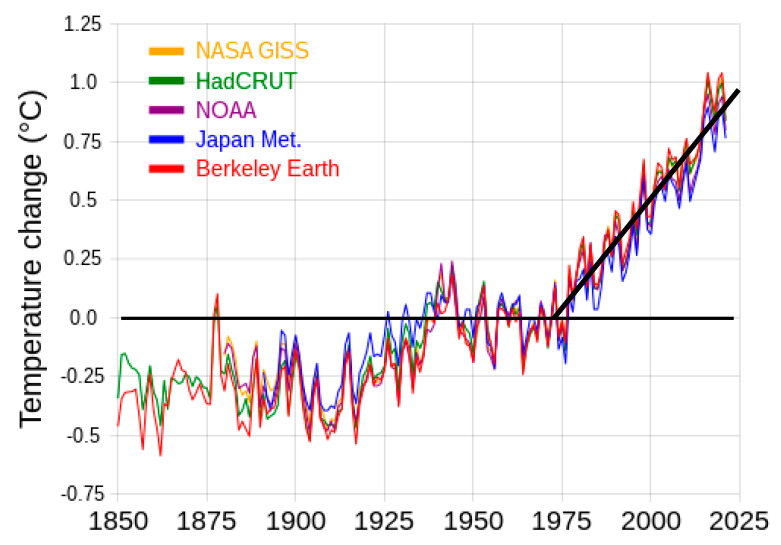
Graphs showing the correlation of the measured global average temperature changes since the 1950’s, from five different scientific organizations: the periodic peaks-troughs with a roughly 4-year frequency are caused by the slight effect of planetary objects in the solar system upon the Earth’s ellipticity and tilt that change the duration of the seasons slightly. The start of global warming relative to the 1950’s coincides with Figure 15 increase in natural gas exploitation. https://commons.wikimedia.org/wiki/File:20200324_Global_average_temperature_-_NASA-GISS_HadCrut_NOAA_Japan_BerkeleyE.svg (accessed on 5 August 2022).

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
