# Peer review of "Global Warming by Geothermal Heat from Fracking: Energy Industry’s Enthalpy Footprints"

_entropy, 2022, doi:10.3390/e24091316_

Round 1
Reviewer 1 Report
Manuscript title:
Global Warming by Geothermal Heat from Fracking: Energy Industry’s Enthalpy Footprints
by: Leslie V. Woodcock
This paper provides a comprehensive analysis by showing the relationship of geothermal heat from fracking with global warming. The second part of the paper discusses the basic principles of the laws of thermodynamics in relation to energy industry’s footprints from the heat generated by global fuel consumption.
I consider this an insightful and enlightening paper that should provide a new perspective about the issue of global warming to those in climate science community as well as general public.
The paper is well written, I recommend the publication of this paper with some suggestions for the author to consider, as follows:
1. The paper would gain an impact if the author could provide an explanation or correlate the role of methane on the whole global warming issue... as fracking operation has also been associated with releasing methane, a much more potent greenhouse gas.
2. Provide reference to paragraph 2 of Fracking and Earthquakes, in particular reference to the statement of ‘… overwhelming evidence that fracking in the vicinity….’
3. There are a few typo errors in page 5 and 13:
i. Page 5, line 13, Section 2.3: …. ‘sea level’, instead of ‘seal level’
ii. Page 5, line 19: ‘… less than 40mW/m2’, instead of ‘40wM/m2’
iii. Page 13, line 12: ‘… liquid octane’, instead of ‘liqiod octane’
iv. Page 13, line 14: ‘… of more than 5000 kJ’, instead of ‘50000 kJ’?
v. Page 13, line 26: ‘… from 1949 to 2019’, instead of ‘1949 to 20119’
Author Response
I have accepted the recommended improvements and corrections of reviewer 1

Reviewer 2 Report
The paper makes the point that that hot spots in global temperatures at the surface of the earth can be explained by enhanced transfer of heat via seismic pathways created as a result of fracking. The author then makes the more general point that the observed increases in global temperatures can be explained based on the enthalpic print of all energy sources and not on the greenhouse effect. In this view there is no "sustainable" energy as all all energy regardless of its source is dissipated in the atmosphere.
This is an interesting hypothesis, even intriguing, but Entropy is not the proper place for this work. Neither the readership nor the reviewing expertise of the journal can properly assess and benefit from this paper. The connection to thermodynamics is secondary to its climate aspect. Heat of reactions, enthalpy and entropy calculations and all other thermodynamic calculations in the paper are standard and correct. It is the implication of the results that must be vetted. I am sure that the climate research community would have legitimate questions to ask, questions that I as an expert on thermodynamics cannot.
My recommendation is that the paper should be submitted to a journal with strong climate focus.
Author Response
I have revised the manuscript to accomodate some of the reservations of this reviewer

Reviewer 3 Report
The paper develops the analysis of the global warming, considering the energy industry’s enthalpy footprints from the heat generated by all fuel consumption,
and the electricity usage enthalpy footprint.
It is very intersting. I can only suggest to improve the link between global warming and the need of sustainable development. I can suggest a recent paper on this topic by Grisolia et al. (Journal of Thermal Analysis and Calorimetry 145, 701-707 (2021) and Sustainability 14, 5679 (2022)).
I must also suggest to use the italic style for the physical quantities (enthalpies, etc.).
Last, related to the Le Chatelier's principle there is also the constructal law approach related to fluxes. The author could consider also this approach.
Author Response
I have included in the revised version the recommendations of reviewer 3 and included the references

Round 2
Reviewer 1 Report
The author has made substantial corrections, I recommend acceptance of the manuscript.
Reviewer 3 Report
I suggest to accept the paper as it is.